# Defense-GAN: Protecting Classifiers Against Adversarial Attacks Using Generative Models

**Pouya Samangouei**[*]**, Maya Kabkab**[*]**, and Rama Chellappa**
Department of Electrical and Computer Engineering
University of Maryland Institute for Advanced Computer Studies
University of Maryland, College Park, MD 20742
`{pouya, mayak, rama}@umiacs.umd.edu`

## Abstract

In recent years, deep neural network approaches have been widely adopted for machine learning tasks, including classification. However, they were shown to be vulnerable to adversarial perturbations: carefully crafted small perturbations can cause misclassification of legitimate images. We propose Defense-GAN, a new framework leveraging the expressive capability of generative models to defend deep neural networks against such attacks. Defense-GAN is trained to model the distribution of unperturbed images. At inference time, it finds a close output to a given image which does not contain the adversarial changes. This output is then fed to the classifier. Our proposed method can be used with any classification model and does not modify the classifier structure or training procedure. It can also be used as a defense against any attack as it does not assume knowledge of the process for generating the adversarial examples. We empirically show that Defense-GAN is consistently effective against different attack methods and improves on existing defense strategies.

## 1 Introduction

Despite their outstanding performance on several machine learning tasks, deep neural networks have been shown to be susceptible to *adversarial attacks* (Szegedy et al., 2014; Goodfellow et al., 2015). These attacks come in the form of *adversarial examples*: carefully crafted perturbations added to a legitimate input sample. In the context of classification, these perturbations cause the legitimate sample to be misclassified at inference time (Szegedy et al., 2014; Goodfellow et al., 2015; Papernot et al., 2016b; Liu et al., 2017). Such perturbations are often small in magnitude and do not affect human recognition but can drastically change the output of the classifier.

Recent literature has considered two types of threat models: *black-box* and *white-box* attacks. Under the black-box attack model, the attacker does not have access to the classification model parameters; whereas in the white-box attack model, the attacker has complete access to the model architecture and parameters, including potential defense mechanisms (Papernot et al., 2017; Tramèr et al., 2017; Carlini & Wagner, 2017).

Various defenses have been proposed to mitigate the effect of adversarial attacks. These defenses can be grouped under three different approaches: (1) modifying the training data to make the classifier more robust against attacks, e.g., *adversarial training* which augments the training data of the classifier with adversarial examples (Szegedy et al., 2014; Goodfellow et al., 2015), (2) modifying the training procedure of the classifier to reduce the magnitude of gradients, e.g., defensive distillation (Papernot et al., 2016d), and (3) attempting to remove the adversarial noise from the input samples (Hendrycks & Gimpel, 2017; Meng & Chen, 2017). All of these approaches have limitations in the sense that they are effective against either white-box attacks or black-box attacks, but not both (Tramèr et al., 2017; Meng & Chen, 2017). Furthermore, some of these defenses are devised with specific attack models in mind and are not effective against new attacks.

---

[*]The first two authors contributed equally.

In this paper, we propose a novel defense mechanism which is effective against both white-box and black-box attacks. We propose to leverage the representative power of Generative Adversarial Networks (GAN) (Goodfellow et al., 2014) to diminish the effect of the adversarial perturbation, by "projecting" input images onto the range of the GAN's generator prior to feeding them to the classifier. In the GAN framework, two models are trained simultaneously in an adversarial setting: a generative model that emulates the data distribution, and a discriminative model that predicts whether a certain input came from real data or was artificially created. The generative model learns a mapping $G$ from a low-dimensional vector $\mathbf{z} \in \mathbb{R}^k$ to the high-dimensional input sample space $\mathbb{R}^n$. During training of the GAN, $G$ is encouraged to generate samples which resemble the training data. It is, therefore, expected that legitimate samples will be close to some point in the range of $G$, whereas adversarial samples will be further away from the range of $G$. Furthermore, "projecting" the adversarial examples onto the range of the generator $G$ can have the desirable effect of reducing the adversarial perturbation. The projected output, computed using Gradient Descent (GD), is fed into the classifier instead of the original (potentially adversarially modified) image. We empirically demonstrate that this is an effective defense against both black-box and white-box attacks on two benchmark image datasets.

The rest of the paper is organized as follows. We introduce the necessary background regarding known attack models, defense mechanisms, and GANs in Section 2. Our defense mechanism, which we call *Defense-GAN*, is formally motivated and introduced in Section 3. Finally, experimental results, under different threat models, as well as comparisons to other defenses are presented in Section 4.

## 2    RELATED WORK AND BACKGROUND INFORMATION

In this work, we propose to use GANs for the purpose of defending against adversarial attacks in classification problems. Before detailing our approach in the next section, we explain related work in three parts. First, we discuss different attack models employed in the literature. We, then, go over related defense mechanisms against these attacks and discuss their strengths and shortcomings. Lastly, we explain necessary background information regarding GANs.

### 2.1    ATTACK MODELS AND ALGORITHMS

Various attack models and algorithms have been used to target classifiers. All attack models we consider aim to find a perturbation $\boldsymbol{\delta}$ to be added to a (legitimate) input $\mathbf{x} \in \mathbb{R}^n$, resulting in the adversarial example $\tilde{\mathbf{x}} = \mathbf{x} + \boldsymbol{\delta}$. The $\ell_\infty$-norm of the perturbation is denoted by $\epsilon$ (Goodfellow et al., 2015) and is chosen to be small enough so as to remain undetectable. We consider two threat levels: black- and white-box attacks.

#### 2.1.1    WHITE-BOX ATTACK MODELS

White-box models assume that the attacker has complete knowledge of all the classifier parameters, i.e., network architecture and weights, as well as the details of any defense mechanism. Given an input image $\mathbf{x}$ and its associated ground-truth label $y$, the attacker thus has access to the loss function $J(\mathbf{x}, y)$ used to train the network, and uses it to compute the adversarial perturbation $\boldsymbol{\delta}$. Attacks can be *targeted*, in that they attempt to cause the perturbed image to be misclassified to a specific target class, or *untargeted* when no target class is specified.

In this work, we focus on untargeted white-box attacks computed using the Fast Gradient Sign Method (FGSM) (Goodfellow et al., 2015), the Randomized Fast Gradient Sign Method (RAND+FGSM) (Tramèr et al., 2017), and the Carlini-Wagner (CW) attack (Carlini & Wagner, 2017). Although other attack models exist, such as the Iterative FGSM (Kurakin et al., 2017), the Jacobian-based Saliency Map Attack (JSMA) (Papernot et al., 2016b), and Deepfool (Moosavi-Dezfooli et al., 2016), we focus on these three models as they cover a good breadth of attack algorthims. FGSM is a very simple and fast attack algorithm which makes it extremely amenable to real-time attack deployment. On the other hand, RAND+FGSM, an equally simple attack, increases the power of FGSM for white-box attacks (Tramèr et al., 2017), and finally, the CW attack is one of the most powerful white-box attacks to-date (Carlini & Wagner, 2017).

**Fast Gradient Sign Method (FGSM)**    Given an image $\mathbf{x}$ and its corresponding true label $y$, the FGSM attack sets the perturbation $\boldsymbol{\delta}$ to:

$$\boldsymbol{\delta} = \epsilon \cdot \text{sign}(\nabla_{\mathbf{x}} J(\mathbf{x}, y)). \tag{1}$$

FGSM (Goodfellow et al., 2015) was designed to be extremely fast rather than optimal. It simply uses the sign of the gradient at every pixel to determine the direction with which to change the corresponding pixel value.

**Randomized Fast Gradient Sign Method (RAND+FGSM)**    The RAND+FGSM (Tramèr et al., 2017) attack is a simple yet effective method to increase the power of FGSM against models which were adversarially trained. The idea is to first apply a small random perturbation before using FGSM. More explicitly, for $\alpha < \epsilon$, random noise is first added to the legitimate image $\mathbf{x}$:

$$\mathbf{x}' = \mathbf{x} + \alpha \cdot \text{sign}(\mathcal{N}(\mathbf{0}^n, \mathbf{I}^n)). \tag{2}$$

Then, the FGSM attack is computed on $\mathbf{x}'$, resulting in

$$\tilde{\mathbf{x}} = \mathbf{x}' + (\epsilon - \alpha) \cdot \text{sign}(\nabla_{\mathbf{x}'} J(\mathbf{x}', y)). \tag{3}$$

**The Carlini-Wagner (CW) attack**    The CW attack is an effective optimization-based attack model (Carlini & Wagner, 2017). In many cases, it can reduce the classifier accuracy to almost $0\%$ (Carlini & Wagner, 2017; Meng & Chen, 2017). The perturbation $\boldsymbol{\delta}$ is found by solving an optimization problem of the form:

$$\min_{\boldsymbol{\delta} \in \mathbb{R}^n} \quad ||\boldsymbol{\delta}||_p + c \cdot f(\mathbf{x} + \boldsymbol{\delta})$$
$$\text{s.t.} \quad \mathbf{x} + \boldsymbol{\delta} \in [0, 1]^n, \tag{4}$$

where $f$ is an objective function that drives the example $\mathbf{x}$ to be misclassified, and $c > 0$ is a suitably chosen constant. The $\ell_2$, $\ell_0$, and $\ell_\infty$ norms are considered. We refer the reader to (Carlini & Wagner, 2017) for details regarding the approach to solving (4) and setting the constant $c$.

### 2.1.2    BLACK-BOX ATTACK MODELS

For black-box attacks we consider untargeted FGSM attacks computed on a substitute model (Papernot et al., 2017). As previously mentioned, black-box adversaries have no access to the classifier or defense parameters. It is further assumed that they do not have access to a large training dataset but can query the targeted DNN as a black-box, i.e., access labels produced by the classifier for specific query images. The adversary trains a model, called *substitute*, which has a (potentially) different architecture than the targeted classifier, using a very small dataset augmented by synthetic images labeled by querying the classifier. Adversarial examples are then found by applying any attack method on the substitute network. It was found that such examples designed to fool the substitute often end up being misclassified by the targeted classifier (Szegedy et al., 2014; Papernot et al., 2017). In other words, black-box attacks are easily transferrable from one model to the other.

## 2.2    DEFENSE MECHANISMS

Various defense mechanisms have been employed to combat the threat from adversarial attacks. In what follows, we describe one representative defense strategy from each of the three general groups of defenses.

### 2.2.1    ADVERSARIAL TRAINING

A popular approach to defend against adversarial noise is to augment the training dataset with adversarial examples (Szegedy et al., 2014; Goodfellow et al., 2015; Moosavi-Dezfooli et al., 2016). Adversarial examples are generated using one or more chosen attack models and added to the training set. This often results in increased robustness when the attack model used to generate the augmented training set is the same as that used by the attacker. However, adversarial training does not perform as well when a different attack strategy is used by the attacker. Additionally, it tends to make the model more robust to white-box attacks than to black-box attacks due to *gradient masking* (Papernot et al., 2016c; 2017; Tramèr et al., 2017).

### 2.2.2 DEFENSIVE DISTILLATION

Defensive distillation (Papernot et al., 2016d) trains the classifier in two rounds using a variant of the distillation (Hinton et al., 2014) method. This has the desirable effect of learning a smoother network and reducing the amplitude of gradients around input points, making it difficult for attackers to generate adversarial examples (Papernot et al., 2016d). It was, however, shown that, while defensive distillation is effective against white-box attacks, it fails to adequately protect against black-box attacks transferred from other networks (Carlini & Wagner, 2017).

### 2.2.3 MAGNET

Recently, Meng & Chen (2017) introduced MagNet as an effective defense strategy. It trains a *reformer* network (which is an auto-encoder or a collection of auto-encoders) to move adversarial examples closer to the manifold of legitimate, or natural, examples. When using a collection of auto-encoders, one reformer network is chosen at random at test time, thus strengthening the defense. It was shown to be an effective defense against gray-box attacks where the attacker knows everything about the network and defense, except the parameters. MagNet is the closest defense to our approach, as it attempts to reform an adversarial sample using a learnt auto-encoder. The main differences between MagNet and our approach are: (1) we use GANs instead of auto-encoders, and, most importantly, (2) we use GD minimization to find latent codes as opposed to a feedforward encoder network. This makes Defense-GAN more robust, especially against white-box attacks.

### 2.3 GENERATIVE ADVERSARIAL NETWORKS (GANS)

GANs, originally introduced by Goodfellow et al. (2014), consist of two neural networks, $G$ and $D$. $G : \mathbb{R}^k \rightarrow \mathbb{R}^n$ maps a low-dimensional latent space to the high dimensional sample space of $\mathbf{x}$. $D$ is a binary neural network classifier. In the training phase, $G$ and $D$ are typically learned in an adversarial fashion using actual input data samples $\mathbf{x}$ and random vectors $\mathbf{z}$. An isotropic Gaussian prior is usually assumed on $\mathbf{z}$. While $G$ learns to generate outputs $G(\mathbf{z})$ that have a distribution similar to that of $\mathbf{x}$, $D$ learns to discriminate between "real" samples $\mathbf{x}$ and "fake" samples $G(\mathbf{z})$. $D$ and $G$ are trained in an alternating fashion to minimize the following min-max loss (Goodfellow et al., 2014):

$$\min_G \max_D V(D, G) = \mathbb{E}_{\mathbf{x} \sim p_{\text{data}}(\mathbf{x})}[\log D(\mathbf{x})] + \mathbb{E}_{\mathbf{z} \sim p_{\mathbf{z}}(\mathbf{z})}[\log(1 - D(G(\mathbf{z})))]. \tag{5}$$

It was shown that the optimal GAN is obtained when the resulting generator distribution $p_g = p_{\text{data}}$ (Goodfellow et al., 2014).

However, GANs turned out to be difficult to train in practice (Gulrajani et al., 2017), and alternative formulations have been proposed. Arjovsky et al. (2017) introduced Wasserstein GANs (WGANs) which are a variant of GANs that use the Wasserstein distance, resulting in a loss function with more desirable properties:

$$\min_G \max_D V_W(D, G) = \mathbb{E}_{\mathbf{x} \sim p_{\text{data}}(\mathbf{x})}[D(\mathbf{x})] - \mathbb{E}_{\mathbf{z} \sim p_{\mathbf{z}}(\mathbf{z})}[D(G(\mathbf{z}))]. \tag{6}$$

In this work, we use WGANs as our generative model due to the stability of their training methods, especially using the approach in (Gulrajani et al., 2017).

## 3 PROPOSED DEFENSE-GAN

We propose a new defense strategy which uses a WGAN trained on legitimate (un-perturbed) training samples to "denoise" adversarial examples. At test time, prior to feeding an image $\mathbf{x}$ to the classifier, we project it onto the range of the generator by minimizing the reconstruction error $||G(\mathbf{z}) - \mathbf{x}||_2^2$, using $L$ steps of GD. The resulting reconstruction $G(\mathbf{z})$ is then given to the classifier. Since the generator was trained to model the unperturbed training data distribution, we expect this added step to result in a substantial reduction of any potential adversarial noise. We formally motivate this approach in the following section.

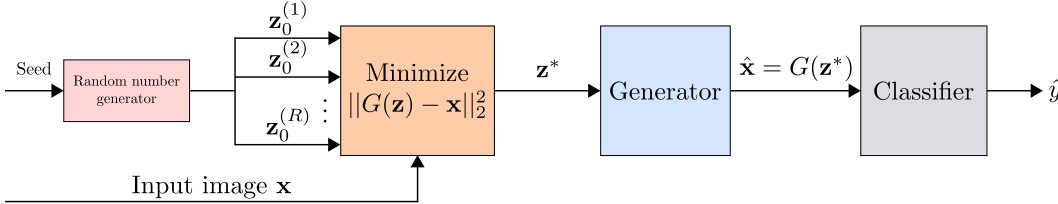

Figure 1: Overview of the Defense-GAN algorithm.

## 3.1 Motivation

As mentioned in Section 2.3, the GAN min-max loss in (5) admits a global optimum when $p_g = p_{\text{data}}$ (Goodfellow et al., 2014). It can be similarly shown that WGAN admits an optimum to its own min-max loss in (6), when the set $\{\mathbf{x} \mid p_g(\mathbf{x}) \neq p_{\text{data}}(\mathbf{x})\}$ has zero Lebesgue-measure. Formally,

**Lemma 1** *A generator distribution $p_g$ is a global optimum for the WGAN min-max game defined in (6), if and only if $p_g(\mathbf{x}) = p_{data}(\mathbf{x})$ for all $\mathbf{x} \in \mathbb{R}^n$, potentially except on a set of zero Lebesgue-measure.*

A sketch of the proof can be found in Appendix A.

Additionally, it was shown that, if $G$ and $D$ have enough capacity to represent the data, and if the training algorithm is such that $p_g$ converges to $p_{\text{data}}$, then

$$\mathbb{E}_{\mathbf{x} \sim p_{\text{data}}} \left[ \min_{\mathbf{z}} ||G_t(\mathbf{z}) - \mathbf{x}||_2 \right] \longrightarrow 0 \tag{7}$$

where $G_t$ is the generator of a GAN or WGAN[1] after $t$ steps of its training algorithm (Kabkab et al., 2018).

This serves to show that, under ideal conditions, the addition of the GAN reconstruction loss minimization step should not affect the performance of the classifier on natural, legitimate samples, as such samples should be almost exactly recovered. Furthermore, we hypothesize that this step will help reduce the adversarial noise which follows a different distribution than that of the GAN training examples.

## 3.2 Defense-GAN algorithm

Defense-GAN is a defense strategy to combat both white-box and black-box adversarial attacks against classification networks. At inference time, given a trained GAN generator $G$ and an image $\mathbf{x}$ to be classified, $\mathbf{z}^*$ is first found so as to minimize

$$\min_{\mathbf{z}} ||G(\mathbf{z}) - \mathbf{x}||_2^2. \tag{8}$$

$G(\mathbf{z}^*)$ is then given as the input to the classifier. The algorithm is illustrated in Figure 1. As (8) is a highly non-convex minimization problem, we approximate it by doing a fixed number $L$ of GD steps using $R$ different random initializations of $\mathbf{z}$ (which we call random restarts), as shown in Figures 1 and 2.

The GAN is trained on the available classifier training dataset in an unsupervised manner. The classifier can be trained on the original training images, their reconstructions using the generator $G$, or a combination of the two. As was discussed in Section 3.1, as long as the GAN is appropriately trained and has enough capacity to represent the data, original clean images and their reconstructions should not defer much. Therefore, these two classifier training strategies should, at least theoretically, not differ in performance.

Compared to existing defense mechanisms, our approach is different in the following aspects:

---

[1]For simplicity, we will use GAN and WGAN interchangeably in the rest of this manuscript, with the understanding that our implementation follows the WGAN loss.

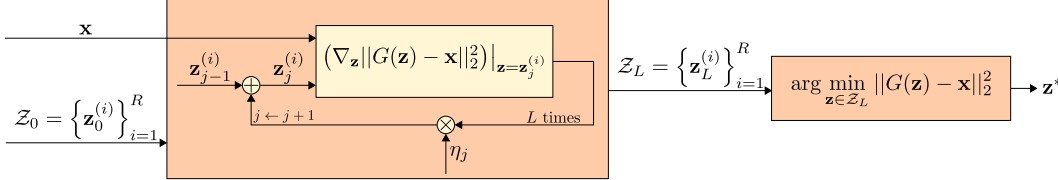

Figure 2: $L$ steps of Gradient Descent are used to estimate the projection of the image onto the range of the generator.

1. Defense-GAN can be used in conjunction with any classifier and does not modify the classifier structure itself. It can be seen as an add-on or pre-processing step prior to classification.

2. If the GAN is representative enough, re-training the classifier should not be necessary and any drop in performance due to the addition of Defense-GAN should not be significant.

3. Defense-GAN can be used as a defense to any attack: it does not assume an attack model, but simply leverages the generative power of GANs to reconstruct adversarial examples.

4. Defense-GAN is highly non-linear and white-box gradient-based attacks will be difficult to perform due to the GD loop. A detailed discussion about this can be found in Appendix B.

## 4 EXPERIMENTS

We assume three different attack threat levels:

1. Black-box attacks: the attacker does not have access to the details of the classifier and defense strategy. It therefore trains a substitute network to find adversarial examples.

2. White-box attacks: the attacker knows all the details of the classifier and defense strategy. It can compute gradients on the classifier and defense networks in order to find adversarial examples.

3. White-box attacks, revisited: in addition to the details of the architectures and parameters of the classifier and defense, the attacker has access to the random seed and random number generator. In the case of Defense-GAN, this means that the attacker knows all the random initializations $\{\mathbf{z}_0^{(i)}\}_{i=1}^R$.

We compare our method to adversarial training (Goodfellow et al., 2015) and MagNet (Meng & Chen, 2017) under the FGSM, RAND+FGSM, and CW (with $\ell_2$ norm) white-box attacks, as well as the FGSM black-box attack. Details of all network architectures used in this paper can be found in Appendix C. When the classifier is trained using the reconstructed images ($G(\mathbf{z}^*)$), we refer to our method as Defense-GAN-Rec, and we use Defense-GAN-Orig when the original images ($\mathbf{x}$) are used to train the classifier. Our GAN follows the WGAN training procedure in (Gulrajani et al., 2017), and details of the generator and discriminator network architectures are given in Table 6. The reformer network (encoder) for the MagNet baseline is provided in Table 7. Our implementation is based on TensorFlow (Abadi et al., 2015) and builds on open-source software: CleverHans by Papernot et al. (2016a) and improved WGAN training by Gulrajani et al. (2017). We use machines equipped with NVIDIA GeForce GTX TITAN X GPUs.

In our experiments, we use two different image datasets: the MNIST handwritten digits dataset (LeCun et al., 1998) and the Fashion-MNIST (F-MNIST) clothing articles dataset (Xiao et al., 2017). Both datasets consist of $60,000$ training images and $10,000$ testing images. We split the training images into a training set of $50,000$ images and hold-out a validation set containing $10,000$ images. For white-box attacks, the testing set is kept the same ($10,000$ samples). For black-box attacks, the testing set is divided into a small hold-out set of $150$ samples reserved for adversary substitute training, as was done in (Papernot et al., 2017), and the remaining $9,850$ samples are used for testing the different methods.

## 4.1 RESULTS ON BLACK-BOX ATTACKS

In this section, we present experimental results on FGSM black-box attacks. As previously mentioned, the attacker trains a substitute model, which could differ in architecture from the targeted model, using a limited dataset consisting of 150 legitimate images augmented with synthetic images labeled using the target classifier. The classifier and substitute model architectures used and referred to throughout this section are described in Table 5 in the Appendix.

In Tables 1 and 2, we present our classification accuracy results and compare to other defense methods. As can be seen, FGSM black-box attacks were successful at reducing the classifier accuracy by up to 70%. All considered defense mechanisms are relatively successful at diminishing the effect of the attacks. We note that, as expected, the performance of Defense-GAN-Rec and that of Defense-GAN-Orig are very close. In addition, they both perform consistently well across different classifier and substitute model combinations. MagNet also performs in a consistent manner, but achieves lower accuracy than Defense-GAN. Two adversarial training defenses are presented: the first one obtains the adversarial examples assuming the same attack $\epsilon = 0.3$, and the second assumes a different $\epsilon = 0.15$. With incorrect knowledge of $\epsilon$, the performance of adversarial training generally decreases. In addition, the classification performance of this defense method has very large variance across the different architectures. It is worth noting that adversarial training defense is only fit against FGSM attacks, because the adversarially augmented data, even with a different $\epsilon$, is generated using the same method as the black-box attack (FGSM). In contrast, Defense-GAN and MagNet are general defense mechanisms which do not assume a specific attack model.

The performances of defenses on the F-MNIST dataset, shown in Table 2, are noticeably lower than on MNIST. This is due to the large $\epsilon = 0.3$ in the FGSM attack. Please see Appendix D for qualitative examples showing that $\epsilon = 0.3$ represents very high noise, which makes F-MNIST images difficult to classify, even by a human.

In addition, the Defense-GAN parameters used in this experiment were kept the same for both Tables, in order to study the effect of dataset complexity, and can be further optimized as investigated in the next section.

Table 1: Classification accuracies of different classifier and substitute model combinations using various defense strategies on the MNIST dataset, under FGSM black-box attacks with $\epsilon = 0.3$. Defense-GAN has $L = 200$ and $R = 10$.

| Classifier/ Substitute | No Attack | No Defense | **Defense-GAN-Rec** | **Defense-GAN-Orig** | MagNet | Adv. Tr. $\epsilon = 0.3$ | Adv. Tr. $\epsilon = 0.15$ |
|---|---|---|---|---|---|---|---|
| A/B | 0.9970 | 0.6343 | 0.9312 | 0.9282 | 0.6937 | **0.9654** | 0.6223 |
| A/E | 0.9970 | 0.5432 | 0.9139 | 0.9221 | 0.6710 | **0.9668** | 0.9327 |
| B/B | 0.9618 | 0.2816 | 0.9057 | **0.9105** | 0.5687 | 0.2092 | 0.3441 |
| B/E | 0.9618 | 0.2128 | 0.8841 | **0.8892** | 0.4627 | 0.1120 | 0.3354 |
| C/B | 0.9959 | 0.6648 | 0.9357 | 0.9322 | 0.7571 | **0.9834** | 0.9208 |
| C/E | 0.9959 | 0.8050 | 0.9223 | 0.9182 | 0.6760 | **0.9843** | 0.9755 |
| D/B | 0.9920 | 0.4641 | 0.9272 | **0.9323** | 0.6817 | 0.7667 | 0.8514 |
| D/E | 0.9920 | 0.3931 | **0.9164** | 0.9155 | 0.6073 | 0.7676 | 0.7129 |

### 4.1.1 EFFECT OF NUMBER OF GD ITERATIONS $L$ AND RANDOM RESTARTS $R$

Figure 3 shows the effect of varying the number of GD iterations $L$ as well as the random restarts $R$ used to compute the GAN reconstructions of input images. Across different $L$ and $R$ values, Defense-GAN-Rec and Defense-GAN-Orig have comparable performance. Increasing $L$ has the expected effect of improving performance when no attack is present. Interestingly, with an FGSM attack, the classification performance decreases after a certain $L$ value. With too many GD iterations on the mean squared error (MSE) $||G(\mathbf{z}) - (\mathbf{x} + \boldsymbol{\delta})||_2^2$, some of the adversarial noise components are retained. In the right Figure, the effect of varying $R$ is shown to be extremely pronounced. This is due to the non-convex nature of the MSE, and increasing $R$ enables us to sample different local minima.

Table 2: Classification accuracies of different classifier and substitute model combinations using various defense strategies on the F-MNIST dataset, under FGSM black-box attacks with $\epsilon = 0.3$. Defense-GAN has $L = 200$ and $R = 10$.

| Classifier/ Substitute | No Attack | No Defense | **Defense-GAN-Rec** | **Defense-GAN-Orig** | MagNet | Adv. Tr. $\epsilon = 0.3$ | Adv. Tr. $\epsilon = 0.15$ |
|---|---|---|---|---|---|---|---|
| A/B | 0.9346 | 0.5131 | 0.586 | 0.5803 | 0.5404 | **0.7393** | 0.6600 |
| A/E | 0.9346 | 0.3653 | 0.4790 | 0.4616 | 0.3311 | **0.6945** | 0.5638 |
| B/B | 0.7470 | 0.4017 | 0.4940 | **0.5530** | 0.3812 | 0.3177 | 0.3560 |
| B/E | 0.7470 | 0.3123 | 0.3720 | **0.4187** | 0.3119 | 0.2617 | 0.2453 |
| C/B | 0.9334 | 0.2635 | 0.5289 | 0.6079 | 0.4664 | **0.7791** | 0.6838 |
| C/E | 0.9334 | 0.2066 | 0.4871 | 0.4625 | 0.3016 | **0.7504** | 0.6655 |
| D/B | 0.8923 | 0.4541 | 0.5779 | 0.5853 | 0.5478 | 0.6172 | **0.6395** |
| D/E | 0.8923 | 0.2543 | 0.4007 | 0.4730 | 0.3396 | **0.5093** | 0.4962 |

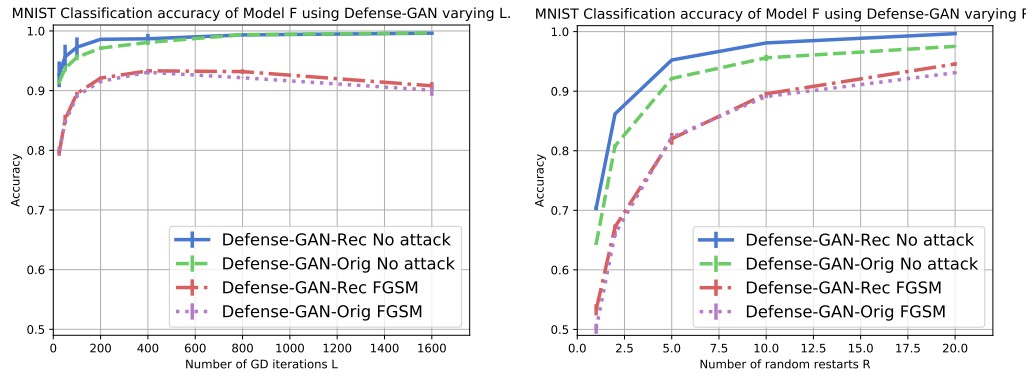

Figure 3: Classification accuracy of Model F using Defense-GAN on the MNIST dataset, under FGSM black-box attacks with $\epsilon = 0.3$ and substitute Model E. Left: various number of iterations $L$ are used ($R = 10$). Right: various number of random restarts $R$ are used ($L = 100$).

### 4.1.2 EFFECT OF ADVERSARIAL NOISE NORM $\epsilon$

We now investigate the effect of changing the attack $\epsilon$ in Table 3. As expected, with higher $\epsilon$, the FGSM attack is more successful, especially on the F-MNIST dataset where the noise norm seems to have a more pronounced effect with nearly 37% drop in performance between $\epsilon = 0.1$ and $0.3$. Figure 7 in Appendix D shows adversarial samples as well as their reconstructions with Defense-GAN at different values of $\epsilon$. We can see that for large $\epsilon$, the class is difficult to discern, even for the human eye.

Even though it seems that increasing $\epsilon$ is a desirable strategy for the attacker, this increases the likelihood that the adversarial noise is discernible and therefore the attack is detected. It is trivial for the attacker to provide adversarial images at very high $\epsilon$, and a good measure of an attack's strength is its ability to affect performance at low $\epsilon$. In fact, in the next section, we discuss how Defense-GAN can be used to not only diminish the effect of attacks, but to also detect them.

### 4.1.3 ATTACK DETECTION

We intuitively expect that clean, unperturbed images will lie closer to the range of the Defense-GAN generator $G$ than adversarial examples. This is due to the fact that $G$ was trained to produce images which resemble the legitimate data. In light of this observation, we propose to use the MSE of an image with it is reconstruction from (8) as a "metric" to decide whether or not the image was

Table 3: Classification accuracy of Model F using Defense-GAN ($L = 400$, $R = 10$), under FGSM black-box attacks for various noise norms $\epsilon$ and substitute Model E.

| $\epsilon$ | Defense-GAN-Rec MNIST | Defense-GAN-Rec F-MNIST |
|---|---|---|
| 0.10 | $0.9864 \pm 0.0011$ | $0.8844 \pm 0.0017$ |
| 0.15 | $0.9836 \pm 0.0026$ | $0.8267 \pm 0.0065$ |
| 0.20 | $0.9772 \pm 0.0019$ | $0.7492 \pm 0.0170$ |
| 0.25 | $0.9641 \pm 0.0001$ | $0.6384 \pm 0.0159$ |
| 0.30 | $0.9307 \pm 0.0034$ | $0.5126 \pm 0.0096$ |

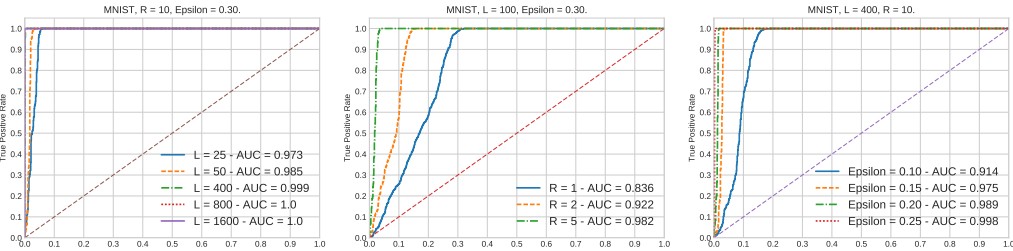

Figure 4: ROC Curves when using Defense-GAN MSE for FGSM attack detections on the MNIST dataset (Classifier Model F, Substitute Model E). Left: Results for various number of GD iterations are shown with $R = 10$, $\epsilon = 0.30$. Middle: Results for various number of random restarts $R$ are shown with $L = 100$, $\epsilon = 0.30$. Right: Results for various $\epsilon$ are shown with $L = 400$, $R = 10$.

adversarially manipulated. In order words, for a given threshold $\theta > 0$, the hypothesis test is:

$$||G(\mathbf{z}^*) - \mathbf{x}||_2^2 \quad \underset{\text{no attack}}{\overset{\text{attack}}{\gtrless}} \quad \theta. \tag{9}$$

We compute the reconstruction MSEs for every image from the test dataset, and its adversarially manipulated version using FGSM. We show the Receiver Operating Characteristic (ROC) curves as well as the Area Under the Curve (AUC) metric for different Defense-GAN parameters and $\epsilon$ values in Figures 4 and 5. The results show that this attack detection strategy is effective especially when the number of GD iterations $L$ and random restarts $R$ are large. From the left and middle Figures, we can conclude that the number of random restarts plays a very important role in the detection false positive and true positive rates as was discussed in Section 4.1.1. Furthermore, when $\epsilon$ is very small, it becomes difficult to detect attacks at low false positive rates.

### 4.1.4 RESULTS ON WHITE-BOX ATTACKS

We now present results on white-box attacks using three different strategies: FGSM, RAND+FGSM, and CW. We perform the CW attack for 100 iterations of projected GD, with learning rate 10.0, and use $c = 100$ in equation (4). Table 4 shows the classification performance of different classifier models across different attack and defense strategies. We note that Defense-GAN significantly outperforms the two other baseline defenses. We even give the adversarial attacker access to the random initializations of $\mathbf{z}$. However, we noticed that the performance does not change much when the attacker does not know the initialization. Adversarial training was done using FGSM to generate the adversarial samples. It is interesting to mention that when CW attack is used, adversarial training performs extremely poorly. As previously discussed, adversarial training does not generalize well against different attack methods.

Due to the loop of $L$ steps of GD, Defense-GAN is resilient to GD-based white-box attacks, since the attacker needs to "un-roll" the GD loop and propagate the gradient of the loss all the way across $L$ steps. In fact, from Table 4, the performance of classifier A with Defense-GAN on the MNIST dataset drops less than $1\%$ from 0.997 to 0.988 under FGSM. In comparison, from Figure 8, when

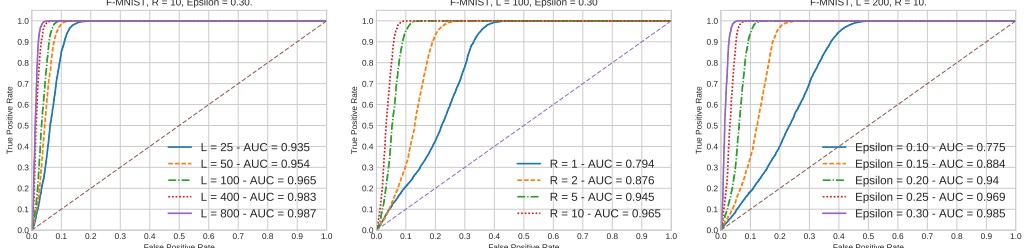

Figure 5: ROC Curves when using Defense-GAN MSE for FGSM attack detections on the F-MNIST dataset (Classifier Model F, Substitute Model E). Left: Results for various number of GD iterations are shown with $R = 10$, $\epsilon = 0.30$. Middle: Results for various number of random restarts $R$ are shown with $L = 100$, $\epsilon = 0.30$. Right: Results for various $\epsilon$ are shown with $L = 200$, $R = 10$.

$L = 25$, the performance of the same network drops to $0.947$ (more than $5\%$ drop). This shows that using a larger $L$ significantly increases the robustness of Defense-GAN against GD-based white-box attacks. This comes at the expense of increased inference time complexity. We present a more detailed discussion about the difficulty of GD-based white-box attacks in Appendix B and time complexity in Appendix G. Additional white-box experimental results on higher-dimensional images are reported in Appendix F.

Table 4: Classification accuracies of different classifier models using various defense strategies on the MNIST (top) and F-MNIST (bottom) datasets, under FGSM, RAND+FGSM, and CW white-box attacks. Defense-GAN has $L = 200$ and $R = 10$.

| Attack | Classifier Model | No Attack | No Defense | **Defense-GAN-Rec** | MagNet | Adv. Tr. $\epsilon = 0.3$ |
|---|---|---|---|---|---|---|
| FGSM $\epsilon = 0.3$ | A | 0.997 | 0.217 | **0.988** | 0.191 | 0.651 |
| | B | 0.962 | 0.022 | **0.956** | 0.082 | 0.060 |
| | C | 0.996 | 0.331 | **0.989** | 0.163 | 0.786 |
| | D | 0.992 | 0.038 | **0.980** | 0.094 | 0.732 |
| RAND+FGSM $\epsilon = 0.3, \alpha = 0.05$ | A | 0.997 | 0.179 | **0.988** | 0.171 | 0.774 |
| | B | 0.962 | 0.017 | **0.944** | 0.091 | 0.138 |
| | C | 0.996 | 0.103 | **0.985** | 0.151 | 0.907 |
| | D | 0.992 | 0.050 | **0.980** | 0.115 | 0.539 |
| CW $\ell_2$ norm | A | 0.997 | 0.141 | **0.989** | 0.038 | 0.077 |
| | B | 0.962 | 0.032 | **0.916** | 0.034 | 0.280 |
| | C | 0.996 | 0.126 | **0.989** | 0.025 | 0.031 |
| | D | 0.992 | 0.032 | **0.983** | 0.021 | 0.010 |

| Attack | Classifier Model | No Attack | No Defense | **Defense-GAN-Rec** | MagNet | Adv. Tr. $\epsilon = 0.3$ |
|---|---|---|---|---|---|---|
| FGSM $\epsilon = 0.3$ | A | 0.934 | 0.102 | **0.879** | 0.089 | 0.797 |
| | B | 0.747 | 0.102 | **0.629** | 0.168 | 0.136 |
| | C | 0.933 | 0.139 | **0.896** | 0.110 | 0.804 |
| | D | 0.892 | 0.082 | **0.875** | 0.099 | 0.698 |
| RAND+FGSM $\epsilon = 0.3, \alpha = 0.05$ | A | 0.934 | 0.102 | **0.888** | 0.096 | 0.447 |
| | B | 0.747 | 0.131 | **0.661** | 0.161 | 0.119 |
| | C | 0.933 | 0.105 | **0.893** | 0.112 | 0.699 |
| | D | 0.892 | 0.091 | **0.862** | 0.104 | 0.626 |
| CW $\ell_2$ norm | A | 0.934 | 0.076 | **0.896** | 0.060 | 0.157 |
| | B | 0.747 | 0.172 | **0.656** | 0.131 | 0.118 |
| | C | 0.933 | 0.063 | **0.896** | 0.084 | 0.107 |
| | D | 0.892 | 0.090 | **0.875** | 0.069 | 0.149 |

## 5 CONCLUSION

In this paper, we proposed Defense-GAN, a novel defense strategy utilizing GANs to enhance the robustness of classification models against black-box and white-box adversarial attacks. Our method does not assume a particular attack model and was shown to be effective against most commonly considered attack strategies. We empirically show that Defense-GAN consistently provides adequate defense on two benchmark computer vision datasets, whereas other methods had many shortcomings on at least one type of attack.

It is worth mentioning that, although Defense-GAN was shown to be a feasible defense mechanism against adversarial attacks, one might come across practical difficulties while implementing and deploying this method. The success of Defense-GAN relies on the expressiveness and generative power of the GAN. However, training GANs is still a challenging task and an active area of research, and if the GAN is not properly trained and tuned, the performance of Defense-GAN will suffer on both original and adversarial examples. Moreover, the choice of hyper-parameters $L$ and $R$ is also critical to the effectiveness of the defense and it may be challenging to tune them without knowledge of the attack.

### ACKNOWLEDGMENT

This research is based upon work supported by the Office of the Director of National Intelligence (ODNI), Intelligence Advanced Research Projects Activity (IARPA), via IARPA R&D Contract No. 2014-14071600012. The views and conclusions contained herein are those of the authors and should not be interpreted as necessarily representing the official policies or endorsements, either expressed or implied, of the ODNI, IARPA, or the U.S. Government. The U.S. Government is authorized to reproduce and distribute reprints for Governmental purposes notwithstanding any copyright annotation thereon.

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

# Appendices

## A  OPTIMALITY OF $p_g = p_{\text{DATA}}$ FOR WGANS

**Sketch of proof of Lemma 1:**  The WGAN min-max loss is given by:

$$V_W(D, G) = \mathbb{E}_{\mathbf{x} \sim p_{\text{data}}(\mathbf{x})}[D(\mathbf{x})] - \mathbb{E}_{\mathbf{z} \sim p_{\mathbf{z}}(\mathbf{z})}[D(G(\mathbf{z}))] \tag{10}$$

$$= \int_{\mathbf{x}} p_{\text{data}}(\mathbf{x})D(\mathbf{x})d\mathbf{x} - \int_{\mathbf{z}} p_{\mathbf{z}}(\mathbf{z})D(G(\mathbf{z}))d\mathbf{z} \tag{11}$$

$$= \int_{\mathbf{x}} \left(p_{\text{data}}(\mathbf{x}) - p_g(\mathbf{x})\right) D(\mathbf{x})d\mathbf{x} \tag{12}$$

For a fixed $G$, the optimal discriminator $D$ which maximizes $V_W(D, G)$ is such that:

$$D_G^*(\mathbf{x}) = \begin{cases} 1 & \text{if } p_{\text{data}}(\mathbf{x}) \geq p_g(\mathbf{x}) \\ 0 & \text{otherwise} \end{cases} \tag{13}$$

Plugging $D_G^*$ back into (12), we get:

$$V_W(D_G^*, G) = \int_{\mathbf{x}} \left(p_{\text{data}}(\mathbf{x}) - p_g(\mathbf{x})\right) D_G^*(\mathbf{x})d\mathbf{x} \tag{14}$$

$$= \int_{\{\mathbf{x} \mid p_{\text{data}}(\mathbf{x}) \geq p_g(\mathbf{x})\}} \left(p_{\text{data}}(\mathbf{x}) - p_g(\mathbf{x})\right) d\mathbf{x} \tag{15}$$

Let $\mathcal{X} = \{\mathbf{x} \mid p_{\text{data}}(\mathbf{x}) \geq p_g(\mathbf{x})\}$. Clearly, to minimize (15), we need to set $p_{\text{data}}(\mathbf{x}) = p_g(\mathbf{x})$ for $\mathbf{x} \in \mathcal{X}$. Then, since both pdfs should integrate to 1,

$$\int_{\mathcal{X}^c} p_g(\mathbf{x})d\mathbf{x} = \int_{\mathcal{X}^c} p_{\text{data}}(\mathbf{x})d\mathbf{x} \tag{16}$$

However, this is a contradiction since $p_g(\mathbf{x}) < p_{\text{data}}(\mathbf{x})$ for $\mathbf{x} \in \mathcal{X}^c$, unless $\mu(\mathcal{X}^c) = 0$ where $\mu$ is the Lebesgue measure. This concludes the proof.

## B  DIFFICULTY OF GD-BASED WHITE-BOX ATTACKS ON DEFENSE-GAN

In order to perform a GD-based white-box attack on models using Defense-GAN, an attacker needs to compute the gradient of the output of the classifier with respect to the input. From Figure 1, the generator and the classifier can be seen as one, combined, feedforward network, through which it is easy to propagate gradients. The difficulty lies in the orange box of the GD optimization detailed in Figure 2.

For the sake of simplicity, let's assume that $R = 1$. Define $\mathcal{L}(\mathbf{x}, \mathbf{z}) = ||G(\mathbf{z}) - \mathbf{x}||_2^2$. Then $\mathbf{z}^* = \mathbf{z}_L$, which is computed recursively as follows:

$$\mathbf{z}_1 = \mathbf{z}_0 + \eta_0 \left. \nabla_{\mathbf{z}}\mathcal{L}(\mathbf{x}, \mathbf{z})\right|_{\mathbf{z}=\mathbf{z}_0} \tag{17}$$

$$\mathbf{z}_2 = \mathbf{z}_1 + \eta_1 \left. \nabla_{\mathbf{z}}\mathcal{L}(\mathbf{x}, \mathbf{z})\right|_{\mathbf{z}=\mathbf{z}_1} \tag{18}$$

$$= \mathbf{z}_0 + \eta_0 \left. \nabla_{\mathbf{z}}\mathcal{L}(\mathbf{x}, \mathbf{z})\right|_{\mathbf{z}=\mathbf{z}_0} + \eta_1 \left. \nabla_{\mathbf{z}}\mathcal{L}(\mathbf{x}, \mathbf{z})\right|_{\mathbf{z}=\mathbf{z}_0+\eta_0 \nabla_{\mathbf{z}}\mathcal{L}(\mathbf{x},\mathbf{z})|_{\mathbf{z}=\mathbf{z}_0}} \tag{19}$$

and so on. Therefore, computing the gradient of $\mathbf{z}^*$ with respect to $\mathbf{x}$ involves a large number ($L$) of recursive chain rules and high-dimensional Jacobian tensors. This computation gets increasingly prohibitive for large $L$.

## C  NEURAL NETWORK ARCHITECTURES

We describe the neural network architectures used throughout the paper. The detail of models A through F used for classifier and substitute networks can be found in Table 5. In Table 6, the GAN architectures are described, and in Table 7, the encoder architecture for the MagNet baseline is given. In what follows:

- Conv($m$, $k \times k$, $s$) refers to a convolutional layer with $m$ feature maps, filter size $k \times k$, and stride $s$

- ConvT($m$, $k \times k$) refers to the transpose (gradient) of Conv (sometimes referred to as "deconvolution") with $m$ feature maps, filter size $k \times k$, and stride $s$

- FC($m$) refers to a fully-connected layer with $m$ outputs

- Dropout($p$) refers to a dropout layer with probability $p$

- ReLU refers to the Rectified Linear Unit activation

- LeakyReLU($\alpha$) is the leaky version of the Rectified Linear Unit with parameter $\alpha$

Table 5: Neural network architectures used for classifiers and substitute models.

| A | B, F* | C | D, E* |
|---|---|---|---|
| Conv(64, $5 \times 5$, 1) | Dropout(0.2) | Conv(128, $3 \times 3$, 1) | FC(200) |
| ReLU | Conv(64, $8 \times 8$, 2) | ReLU | ReLU |
| Conv(64, $5 \times 5$, 2) | ReLU | Conv(64, $3 \times 3$, 2) | Dropout(0.5) |
| ReLU | Conv(128, $6 \times 6$, 2) | ReLU | FC(200) |
| Dropout(0.25) | ReLU | Dropout(0.25) | ReLU |
| FC(128) | Conv(128, $5 \times 5$, 1) | FC(128) | Dropout(0.5) |
| ReLU | ReLU | ReLU | FC(10) + Softmax |
| Dropout(0.5) | Dropout(0.5) | Dropout(0.5) | |
| FC(10) + Softmax | FC(10) + Softmax | FC(10) + Softmax | |

[ * : F (resp. E) shares the same architecture as B (resp. D) with the dropout layers removed ]

Table 6: Neural network architectures used for GANs.

| Generator | Discriminator |
|---|---|
| FC(4096) | Conv(64, $5 \times 5$, 2) |
| ReLU | LeakyReLU(0.2) |
| ConvT(256, $5 \times 5$, 1) | Conv(128, $5 \times 5$, 2) |
| ReLU | LeakyReLU(0.2) |
| ConvT(128, $5 \times 5$, 1) | Conv(256, $5 \times 5$, 2) |
| ReLU | LeakyReLU(0.2) |
| ConvT(1, $5 \times 5$, 1) | FC(1) |
| Sigmoid | Sigmoid |

Table 7: Neural network architecture used for the MagNet encoder.

| Encoder |
|---|
| Conv(64, $5 \times 5$, 2) |
| LeakyReLU(0.2) |
| Conv(128, $5 \times 5$, 2) |
| LeakyReLU(0.2) |
| Conv(256, $5 \times 5$, 2) |
| LeakyReLU(0.2) |
| FC(128) + $\tanh$ |

# D QUALITATIVE EXAMPLES

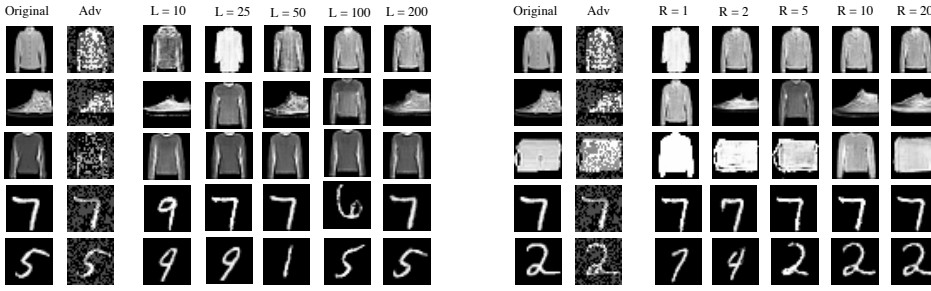

Figure 6: Examples from MNIST and F-MNIST. Left: Original, FGSM adversarial $\epsilon = 0.3$, and reconstruction images for $R = 1$ and various $L$ are shown. Right: Original, FGSM adversarial $\epsilon = 0.3$, and reconstruction images for $L = 25$ and various $R$ are shown.

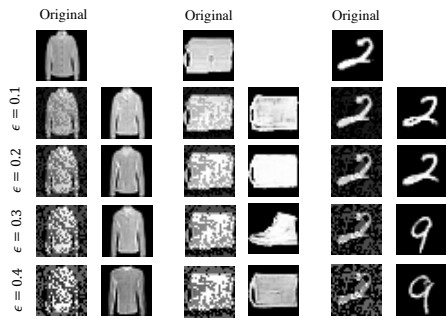

Figure 7: Examples from MNIST and F-MNIST: Original, FGSM adversarial and reconstruction images for $L = 50$, $R = 15$ and various $\epsilon$ are shown.

# E ADDITIONAL RESULTS ON THE EFFECT OF VARYING THE NUMBER OF GD ITERATIONS $L$ AND RANDOM RESTARTS $R$

Table 8: Classification accuracy of Model F using Defense-GAN with various number of iterations $L$ ($R = 10$), on the MNIST dataset, under FGSM black-box attack with $\epsilon = 0.3$.

| $L$ | Defense-GAN-Rec No attack | Defense-GAN-Orig No attack | Defense-GAN-Rec Adversarial | Defense-GAN-Orig Adversarial |
|---|---|---|---|---|
| 25 | $0.9273 \pm 0.0215$ | $0.9141 \pm 0.0033$ | $0.7955 \pm 0.0045$ | $0.7998 \pm 0.0063$ |
| 50 | $0.9567 \pm 0.0203$ | $0.9371 \pm 0.0048$ | $0.8516 \pm 0.0078$ | $0.8472 \pm 0.0026$ |
| 100 | $0.9728 \pm 0.0164$ | $0.9560 \pm 0.0051$ | $0.8953 \pm 0.0027$ | $0.8911 \pm 0.0024$ |
| 200 | $0.9860 \pm 0.0010$ | $0.9712 \pm 0.0028$ | $0.9210 \pm 0.0023$ | $0.9155 \pm 0.0032$ |
| 400 | $0.9869 \pm 0.0082$ | $0.9808 \pm 0.0044$ | $0.9332 \pm 0.0027$ | $0.9307 \pm 0.0034$ |
| 800 | $0.9934 \pm 0.0009$ | $0.9938 \pm 0.0004$ | $0.9319 \pm 0.0038$ | $0.9216 \pm 0.0005$ |
| 1600 | $0.9963 \pm 0.0013$ | $0.9967 \pm 0.0005$ | $0.9081 \pm 0.0062$ | $0.9008 \pm 0.0095$ |

Table 9: Classification accuracy of Model F using Defense-GAN with various number of iterations $L$ ($R = 10$), on the F-MNIST dataset, under FGSM black-box attack with $\epsilon = 0.3$.

| $L$ | Defense-GAN-Rec No attack | Defense-GAN-Orig No attack | Defense-GAN-Rec Adversarial | Defense-GAN-Orig Adversarial |
|---|---|---|---|---|
| 25 | $0.8037 \pm 0.0050$ | $0.7595 \pm 0.0009$ | $0.4040 \pm 0.0149$ | $0.3910 \pm 0.0119$ |
| 50 | $0.8676 \pm 0.0018$ | $0.7898 \pm 0.0016$ | $0.4412 \pm 0.0023$ | $0.3980 \pm 0.0114$ |
| 100 | $0.9101 \pm 0.0032$ | $0.8190 \pm 0.0043$ | $0.4808 \pm 0.0088$ | $0.4221 \pm 0.0255$ |
| 200 | $0.9145 \pm 0.0014$ | $0.8373 \pm 0.0054$ | $0.5119 \pm 0.0038$ | $0.4594 \pm 0.0056$ |
| 400 | $0.9490 \pm 0.0013$ | $0.8557 \pm 0.0049$ | $0.5126 \pm 0.0096$ | $0.4754 \pm 0.0102$ |
| 800 | $0.9588 \pm 0.0065$ | $0.8832 \pm 0.0042$ | $0.5520 \pm 0.0098$ | $0.4644 \pm 0.0092$ |
| 1600 | $0.9640 \pm 0.0010$ | $0.9125 \pm 0.0040$ | $0.5335 \pm 0.0226$ | $0.4952 \pm 0.0155$ |

Table 10: Classification accuracy of Model F using Defense-GAN with various number of random restarts $R$ ($L = 100$), on the MNIST dataset, under FGSM black-box attack with $\epsilon = 0.3$.

| $R$ | Defense-GAN-Rec No attack | Defense-GAN-Orig No attack | Defense-GAN-Rec Adversarial | Defense-GAN-Orig Adversarial |
|---|---|---|---|---|
| 1 | $0.7035 \pm 0.0035$ | $0.6436 \pm 0.0017$ | $0.5329 \pm 0.0094$ | $0.5011 \pm 0.0085$ |
| 2 | $0.8619 \pm 0.0010$ | $0.8080 \pm 0.0029$ | $0.6722 \pm 0.0041$ | $0.6605 \pm 0.0050$ |
| 5 | $0.9523 \pm 0.0006$ | $0.9213 \pm 0.0024$ | $0.8199 \pm 0.0097$ | $0.8228 \pm 0.0038$ |
| 10 | $0.9810 \pm 0.0015$ | $0.9560 \pm 0.0051$ | $0.8956 \pm 0.0032$ | $0.8911 \pm 0.0024$ |
| 20 | $0.9966 \pm 0.0009$ | $0.9753 \pm 0.0010$ | $0.9456 \pm 0.0031$ | $0.9310 \pm 0.0023$ |

Table 11: Classification accuracy of Model F using Defense-GAN with various number of random restarts $R$ ($L = 100$), on the F-MNIST dataset, under FGSM black-box attack with $\epsilon = 0.3$.

| $R$ | Defense-GAN-Rec No attack | Defense-GAN-Orig No attack | Defense-GAN-Rec Adversarial | Defense-GAN-Orig Adversarial |
|---|---|---|---|---|
| 1 | $0.8425 \pm 0.0008$ | $0.5597 \pm 0.0015$ | $0.3504 \pm 0.0102$ | $0.3380 \pm 0.0043$ |
| 2 | $0.8994 \pm 0.0051$ | $0.7793 \pm 0.0023$ | $0.4050 \pm 0.0148$ | $0.3508 \pm 0.0167$ |
| 5 | $0.9260 \pm 0.0028$ | $0.6726 \pm 0.0006$ | $0.4521 \pm 0.0177$ | $0.4024 \pm 0.0085$ |
| 10 | $0.9101 \pm 0.0032$ | $0.8190 \pm 0.0043$ | $0.4808 \pm 0.0088$ | $0.4221 \pm 0.0255$ |

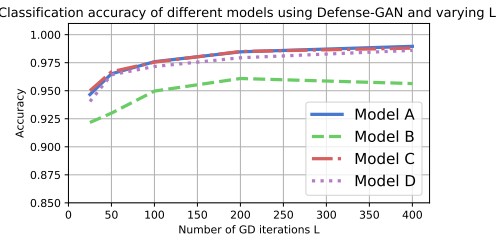

Figure 8: Classification accuracy of different models using Defense-GAN on the MNIST dataset, under FGSM white-box attack with $\epsilon = 0.3$, for various number of iterations $L$ and $R = 10$.

## F    ADDITIONAL RESULTS ON WHITE-BOX ATTACKS

We report results on white-box attacks on the CelebFaces Attributes dataset (CelebA) (Liu et al., 2015) in Table 12. The CelebA dataset is a large-scale face dataset consisting of more than $200,000$ face images, split into training, validation, and testing sets. The RGB images were center-cropped

and resized to $64 \times 64$. We performed the task of gender classification on this dataset. The GAN architecture is the same as that in Table 6, except for an additional ConvT($128, 5 \times 5, 1$) layer in the generator network.

Table 12: Classification accuracies of different classifier models using various defense strategies on the CelebA gender classification task, under FGSM, RAND+FGSM, and CW white-box attacks. Defense-GAN has $L = 200$ and $R = 2$.

| Attack | Classifier Model | No Attack | No Defense | **Defense-GAN-Rec** | MagNet | Adv. Tr. $\epsilon = 0.3$ |
|---|---|---|---|---|---|---|
| FGSM $\epsilon = 0.3$ | A | 0.9652 | 0.0870 | **0.9255** | 0.0985 | 0.1225 |
| | B | 0.9468 | 0.0995 | **0.9140** | 0.0920 | 0.2345 |
| | C | 0.9459 | 0.0460 | **0.9255** | 0.1085 | 0.1130 |
| | D | 0.9476 | 0.0605 | **0.9205** | 0.0975 | 0.7755 |
| RAND+FGSM $\epsilon = 0.3, \alpha = 0.05$ | A | 0.9652 | 0.0560 | **0.9280** | 0.1105 | 0.0700 |
| | B | 0.9468 | 0.1785 | **0.9030** | 0.1015 | 0.4515 |
| | C | 0.9459 | 0.0470 | **0.9200** | 0.1045 | 0.1055 |
| | D | 0.9476 | 0.0665 | **0.9165** | 0.1105 | 0.696 |
| CW $\ell_2$ norm | A | 0.9652 | 0.0460 | **0.8210** | 0.0985 | 0.5690 |
| | B | 0.9468 | 0.0575 | **0.7465** | 0.0955 | 0.0725 |
| | C | 0.9459 | 0.0435 | **0.7985** | 0.0985 | 0.2635 |
| | D | 0.9476 | 0.0660 | **0.7740** | 0.1040 | 0.5010 |

## G  TIME COMPLEXITY

The computational complexity of reconstructing an image using Defense-GAN is on the order of the number of GD iterations performed to estimate $\mathbf{z}^*$, multiplied by the time to compute gradients. The number of random restarts $R$ has less effect on the running time, since random restarts are independent and can run in parallel if enough resources are available. Table 13 shows the average running time, in seconds, to find the reconstructions of MNIST and F-MNIST images on one NVIDIA GeForce GTX TITAN X GPU. For most applications, these running times are not prohibitive. We can see a tradeoff between running time and defense robustness as well as accuracy.

Table 13: Average time, in seconds, to compute reconstructions of MNIST/F-MNIST images for various values of $L$ and $R$.

| | $L = 10$ | $L = 25$ | $L = 50$ | $L = 100$ | $L = 200$ |
|---|---|---|---|---|---|
| $R = 1$ | $0.043 \pm 0.027$ | $0.070 \pm 0.003$ | $0.137 \pm 0.004$ | $0.273 \pm 0.006$ | $L = 0.543 \pm 0.017$ |
| $R = 2$ | $0.042 \pm 0.026$ | $0.067 \pm 0.002$ | $0.131 \pm 0.003$ | $0.261 \pm 0.006$ | $L = 0.510 \pm 0.006$ |
| $R = 5$ | $0.043 \pm 0.029$ | $0.070 \pm 0.002$ | $0.136 \pm 0.004$ | $0.270 \pm 0.004$ | $L = 0.535 \pm 0.008$ |
| $R = 10$ | $0.051 \pm 0.032$ | $0.086 \pm 0.001$ | $0.170 \pm 0.002$ | $0.338 \pm 0.008$ | $L = 0.675 \pm 0.016$ |
| $R = 20$ | $0.060 \pm 0.035$ | $0.105 \pm 0.003$ | $0.209 \pm 0.006$ | $0.414 \pm 0.012$ | $L = 0.825 \pm 0.022$ |

