# OpenReview forum: "Defense-GAN: Protecting Classifiers Against Adversarial Attacks Using Generative Models"
_ICLR.cc/2018/Conference — Accept (Poster)_

### Official Review · AnonReviewer1 · 2017-11-25

**Rating:** 6
**Confidence:** 3

**Review:**

This paper presents a method to cope with adversarial examples in classification tasks, leveraging a generative model of the inputs.  Given an accurate generative model of the input, this approach first projects the input onto the manifold learned by the generative model (the idea being that inputs on this manifold reflect the non-adversarial input distribution).  This projected input is then used to produce the classification probabilities.  The authors test their method on various adversarially constructed inputs (with varying degrees of noise).

Questions/Comments:

- I am interested in unpacking the improvement of Defense-GAN over the MagNet auto-encoder based method.  Is the MagNet auto-encoder suffering lower accuracy because the projection of an adversarial image is based on an encoding function that is learned only on true data?  If the decoder from the MagNet approach were treated purely as a generative model, and the same optimization-based projection approach (proposed in this work) was followed, would the results be comparable?

- Is there anything special about the GAN approach, versus other generative approaches?

- In the black-box vs. white-box scenarios, can the attacker know the GAN parameters?  Is that what is meant by the "defense network" (in experiments bullet 2)?

- How computationally expensive is this approach take compared to MagNet or other adversarial approaches?

Quality: The method appears to be technically correct.

Clarity: This paper clearly written; both method and experiments are presented well.

Originality: I am not familiar enough with adversarial learning to assess the novelty of this approach.

Significance: I believe the main contribution of this method is the optimization-based approach to project onto a generative model's manifold.  I think this kernel has the potential to be explored further (e.g. computational speed-up, projection metrics).

---

> ### Author Response · Authors · 2017-12-31
> **Answer to AnonReviewer1**
>
> We thank the reviewer for the insightful comments and discussions.
>
> A) Defense-GAN vs. MagNet vs. other generative approaches:
> We believe that the MagNet auto-encoder suffers lower accuracy compared to Defense-GAN due to the fact that the “reconstruction” step in MagNet is a feed-forward network as opposed to an optimization-based projection as in Defense-GAN. Overall, the combination of MagNet and the classifier can be seen as one deeper classification network, and has a wide attack surface compared to Defense-GAN.
> As suggested by the reviewer, if the MagNet decoder (or another generative approach) was treated as a generative model, and the same optimization-based projection approach was followed, the model with more representative power would perform better. From our experience, GANs tend to have more representative power, but this is still an active area of research and discussion. We believe that, since GANs are specifically designed to optimize for generative tasks, using a GAN in conjunction with our proposed optimization-based projection would outperform an encoder with the same projection method. However, this would be an interesting future research direction. In addition, we were able to show some theoretical guarantees regarding the use and representative power of GANs in equation (7).
>
> B) Black- and white-box attacks:
> In our work and previous literature, it is assumed that in black-box scenarios the attacker does not know the classifier network nor the defense mechanism (and any parameters thereof). The only information the attacker can use is the classifier output.
> In white-box scenarios, the attacker knows the entire system including the classifier network, defense mechanisms, and all parameters (which in our case, include GAN parameters). By “defense network” in Experiments bullet 2, we mean the generator network.
>
> C) Computational complexity:
> Defense-GAN adds inference-time complexity to the classifier. As discussed in Appendix G (Appendix F in the original version of the paper), this complexity depends on L, the number of GD steps used to reconstruct images, and (to a lesser extent) R, the number of random restarts. At training time, Defense-GAN requires training a GAN, but no retraining of the classifier is necessary.
> In comparison, MagNet also adds inference-time complexity. However, the time overhead is much smaller than Defense-GAN as MagNet is simply a feedforward network. At training time, the overhead is similar to Defense-GAN (training the encoder, no retraining of the classifier).
> Adversarial training adds no inference-time complexity. However, training time can be significantly larger than for other methods since re-training the classifier is required (preceded by generating the adversarial examples to augment the training dataset).

---

### Official Review · AnonReviewer2 · 2017-11-27
**Interesting but hard to conclude decisively from the current experiments**

**Rating:** 6
**Confidence:** 3

**Review:**

This paper presents Defense-GAN: a GAN that used at test time to map the input generate an image (G(z)) close (in MSE(G(z), x)) to the input image (x), by applying several steps of gradient descent of this MSE. The GAN is a WGAN trained on the train set (only to keep the generator). The goal of the whole approach is to be robust to adversarial examples, without having to change the (downstream task) classifier, only swapping in the G(z) for the x.

+ The paper is easy to follow.
+ It seems (but I am not an expert in adversarial examples) to cite the relevant litterature (that I know of) and compare to reasonably established attacks and defenses.
+ Simple/directly applicable approach that seems to work experimentally, but
- A missing baseline is to take the nearest neighbour of the (perturbed) x from the training set.
- Only MNIST-sized images, and MNIST-like (60k train set, 10 labels) datasets: MNIST and F-MNIST.
- Between 0.043sec and 0.825 sec to reconstruct an MNIST-sized image.
? MagNet results were very often worse than no defense in Table 4, could you comment on that?
- In white-box attacks, it seems to me like L steps of gradient descent on MSE(G(z), x) should be directly extended to L steps of (at least) FGSM-based attacks, at least as a control.

---

> ### Author Response · Authors · 2017-12-31
> **Answer to AnonReviewer2**
>
> We appreciate the constructive criticism and detailed analysis of our paper.
>
> A) Nearest-neighbor baseline:
> Taking the nearest neighbor of the potentially perturbed x from the training set can be seen as a simple way of removing adversarial noise, and is tantamount to a 1-nearest-neighbor (1-NN) classifier. On MNIST, a 1-NN classifier achieves an 88.6% accuracy on FGSM adversarial examples with epsilon = 0.3, found using the B substitute network. Defense-GAN-Rec and Defense-GAN-Orig average about 92.5% across the four different classifier networks when the substitute model is fixed to B. Similar trends are found for other substitute models. There is an improvement of about 4% by using Defense-GAN. It is also worth noting that in the case of MNIST, a 1-NN classifier works reasonably well (achieving around 95% on clean images). This is not the case for more complex datasets: for example, if the problem at hand is face attributes classification, nearest neighbors may not necessarily belong to the same class, and therefore NN classifiers will perform poorly.
>
> B) Only MNIST-sized images:
> Based on the reviewer’s suggestion, we have added additional white-box results on the Large-scale CelebFaces Attributes (CelebA) dataset in the appendix of the paper. The results show that Defense-GAN can still be used with more complex datasets including larger and RGB images. For further details, please refer to Appendix F in the revised version.
>
> C) Time to reconstruct images:
> We agree with the reviewer that Defense-GAN introduces additional inference time by reconstructing images using GD on the MSE loss. However, we show its effectiveness against various attacks, especially in comparison to other simpler defenses. Furthermore, we have not optimized the running time of our algorithm, as it was not the focus of this work. This is a worthwhile effort to pursue in the future by trying to better utilize computational resources.
> Per the reviewer’s comment, we have timed some reconstruction steps for CelebA images (which are 15.6 times larger than MNIST/F-MNIST). For R = 2, we have:
> L = 10, 0.132 sec
> L = 25, 0.106 sec
> L = 50, 0.210 sec
> L = 100, 0.413 sec
> L = 200, 0.824 sec
> The reconstruction time for CelebA did not scale with the size of the image.
>
> D) MagNet results are sometimes worse than no defense in Table 4:
> Even though it seems counter-intuitive that a defense mechanism can sometimes cause a decrease in performance, this stems from the fact that white-box attackers also know the exact defense mechanism used. In the case of MagNet, the defense mechanism is another feedforward network which, in conjunction with the original classifier, can be viewed as a new deeper feedforward network. Attacks on this bigger network can sometimes be more successful than attacks on the original network. Furthermore, MagNet was not designed to be robust against white-box attacks.
>
> E) Using L steps of white-box FGSM:
> Per our understanding, the reviewer is suggesting using iterative FGSM. We do agree that for a fair comparison, L steps of iterative FGSM could be used. However, we note that CW is an iterative optimization-based attack, and is more powerful than iterative FGSM. Since we have shown robustness against CW attacks in Table 4, we believe iterative FGSM results will be similar.

---

> > ### Comment · AnonReviewer2 · 2018-01-23
> > **changed my 5 into 6**
> >
> > B) C) Thanks for the additional experiments, I think they make the paper stronger. In particular they validate that scaling is proportional to L but not (linear in) to image size, and that the method works in RGB.
> > D) OK.
> > A) E) I still think that these additional experiments would help, but I am now marginally convinced that the authors expectations are correct.

---

### Official Review · AnonReviewer3 · 2017-11-28
**A novel idea with room for future work.**

**Rating:** 8
**Confidence:** 4

**Review:**

The authors describe a new defense mechanism against adversarial attacks on classifiers (e.g., FGSM). They propose utilizing Generative Adversarial Networks (GAN), which are usually used for training generative models for an unknown distribution, but have a natural adversarial interpretation. In particular, a GAN consists of a generator NN G which maps a random vector z to an example x, and a discriminator NN D which seeks to discriminate between an examples produced by G and examples drawn from the true distribution. The GAN is trained to minimize the max min loss of D on this discrimination task, thereby producing a G (in the limit) whose outputs are indistinguishable from the true distribution by the best discriminator.

Utilizing a trained GAN, the authors propose the following defense at inference time. Given a sample x (which has been adversarially perturbed), first project x onto the range of G by solving the minimization problem z* = argmin_z ||G(z) - x||_2. This is done by SGD. Then apply any classifier trained on the true distribution on the resulting x* = G(z*).

In the case of existing black-box attacks, the authors argue (convincingly) that the method is both flexible and empirically effective. In particular, the defense can be applied in conjunction with any classifier (including already hardened classifiers), and does not assume any specific attack model. Nevertheless, it appears to be effective against FGSM attacks, and competitive with adversarial training specifically to defend against FGSM.

The authors provide less-convincing evidence that the defense is effective against white-box attacks. In particular, the method is shown to be robust against FGSM, RAND+FGSM, and CW white-box attacks. However, it is not clear to me that the method is invulnerable to novel white-box attacks. In particular, it seems that the attacker can design an x which projects onto some desired x* (using some other method entirely), which then fools the classifier downstream.

Nevertheless, the method is shown to be an effective tool for hardening any classifier against existing black-box attacks
(which is arguably of great practical value). It is novel and should generate further research with respect to understanding its vulnerabilities more completely.

Minor Comments:
The sentence starting “Unless otherwise specified…” at the top of page 7 is confusing given the actual contents of Tables 1 and 2, which are clarified only by looking at Table 5 in the appendix. This should be fixed.

---

> ### Author Response · Authors · 2017-12-31
> **Answer to AnonReviewer3**
>
> We thank the reviewer for the constructive review and comments.
>
> A) Regarding the effectiveness against white-box attacks:
> As the reviewer has pointed out, we have shown the robustness of our method to existing white-box attacks such as FGSM, RAND+FGSM, and CW. Indeed, a good attack strategy could be to design an x which projects onto a desired x* = G(z*). However, this requires solving for:
>
> Find x s.t. the output of the gradient-descent block is z*.
>
> Per our understanding, the reviewer’s suggestion is the following:
> Find a desired x* in the range of the generator which fools the classifier.
> Find an x which projects onto x*, i.e., such that the output of the GD block is z*, where G(z*) = x*.
> Step 1 is a more challenging version of existing attacks, due to the constraint that the adversarial example should lie in the range of the generator. While step 1 could potentially be solvable, the real difficulty lies in step 2. In fact, it is not obvious how to find such an x given x*. What comes to mind is attempting to solve step 2 using an optimization framework, e.g.:
> Minimize (over x, z*) 1
> Subject to G(z*) = x*
>                  z* is the output of the GD block after L steps.
>
> We have shown in Appendix B that solving this problem using GD gets more and more prohibitive as L increases.
> Furthermore, since we use random initializations of z, if the random seed is not accessible by the attacker, there is no guarantee that a fixed x will result in the same fixed z every time after L steps of GD on the MSE.
> Due to these factors, we believe that our method is robust to a wide range of gradient-based white-box attacks. However, we are very much interested in further research of novel attack methods.
>
> B) We have fixed the minor comments by specifically mentioning the classifier and substitute models for every Table and Figure throughout the paper.

---

### Public Comment · (anonymous) · 2017-11-06
**Testing on Datasets Other than MNIST/Adversarial Examples of Generator**

Have you tested your method on other datasets? I wonder if it works with datasets such as CIFAR.

Moreover, it's not clear whether this method can defend against existing attacks, without introducing new vulnerabilities. Here are some possible new attack methods:

1- The generator can certainly output examples that are adversarial for the classifier. Hence, the attacker only needs to find out such examples and perturb the input image to make it similar to them.

2- The attacker can target the minimization block, which uses "L steps of Gradient Descent." By forcing it to output a wrong set of Z_L, the rest of the algorithm (combination of generator/classifier) becomes ineffective, i.e., the minimization block can be the bottleneck.

3- The algorithm takes as input a seed, along with the image. Since for a given seed, the random number generator is deterministic, the attacker can test different seeds and use the one for which the algorithm fails. This attack may work even without perturbing the image.

---

> ### Author Response · Authors · 2017-12-31
> **Answer to anonymous commenter**
>
> We thank the anonymous commenter.
> We have added some additional results on the CelebA dataset in Appendix F.
> Regarding the suggested new attack methods, we note that:
> 1- We believe that this same exact point was raised by AnonReviewer3, and we kindly refer the commenter to part A of our reply to AnonReviewer3.
> 2- It is not clear to us how to “output a wrong set of Z_L” and how to find an input x that will meet this criterion.
> (If by “output a wrong set of Z_L” the reviewer means to inject adversarial noise directly on the set of Z_L, then the attacker has gained access and infiltrated an intermediate step of the system and might as well directly modify the classifier output. This type of attacks was never considered in this literature).
> 3- We believe that the commenter mistakenly assumes the seed to be an external input accessible to and modifiable by the attacker. Even though, in Figure 1, the seed is depicted as input to the system, it is never assumed that the attacker can modify the random seed.

---

### Public Comment · (anonymous) · 2017-12-02
**CW is an optimization based attack**

In your appendix you claim the combined model is hard to attack, but I suspect that might not be the case.

1. CW is an optimization based attack.

2. If you just set up the CW optimization attack, and find some local minima for z* that corresponds to an adversarial attack -- I suspect it might be pretty close to the z* you converge on after a few steps of GD. Perhaps worth a shot trying to just combine the two models and add ||G(z)-x|| as another term in the optimization objective. I suspect CW would work pretty well then.

minimize CW loss function + 0.1*||z*-x||

subject y=f(x)
              z*=G(z) or something like this.

---

> ### Author Response · Authors · 2017-12-31
> **Answer to anonymous commenter**
>
> We thank the anonymous commenter.
> We have modified the title of Appendix B to reflect our claim that attacks based on gradient-descent are difficult to perform.
> Regarding the modified CW optimization attack, our understanding is that the commenter is suggesting the following:
>
> Minimize (over x*, z*)  CW loss(x, x*, G(z*)) + 0.1 ||G(z*) - x*||
>
> First of all, this problem is significantly more difficult to solve than the original CW formulation due to the dependence on x, x*, and G(z*).
> Second, this formulation does not guarantee that when x* is input to the system, z* will be the output of the GD block, and an example “close” to an adversarial example is not necessarily adversarial itself.
> Lastly, the random initialization of z in the GD block serves to add robustness and change the output every time.
>
> All in all, we are extremely interested in further investigating new attack strategies as Defense-GAN was shown to be robust to existing attack models.

---

### Public Comment · (anonymous) · 2017-12-14
**Please use some meaningful attacks!!**

https://arxiv.org/pdf/1711.08478.pdf
Instead of doing gradient descent, it might just help to attack directly.
See how easily APE-GAN cracks!!!!

---

> ### Author Response · Authors · 2017-12-31
> **Answer to anonymous commenter**
>
> We thank the anonymous commenter. Due to the recentness of the paper referred to by the commenter, we have not had the time to analyze it in detail. However, as noted in the paper (page 3), the attacks are actually generated using gradient descent as is the case in all attacks used in our paper.
> The mechanism considered in APE-GAN and that considered in our paper are very different. While MagNet and APE-GAN use a feedforward architecture for their “reconstruction” step, Defense-GAN employs an optimization based projection onto the range of the generator, which holds a good representation of the true data.

---

### Public Comment · (anonymous) · 2017-12-20
**What would this defense do on the concentric spheres dataset?**

This paper shows that models trained on a synthetic dataset are vulnerable to small adversarial perturbations which lie on the data manifold. Thus at least for this dataset it seems like a perfect generator would not perturb the adversarial example at all. Can the authors comment what their proposed defense would do to fix these adversarial examples?

https://openreview.net/forum?id=SyUkxxZ0b

---

> ### Author Response · Authors · 2017-12-31
> **Answer to anonymous commenter**
>
> We thank the anonymous commenter. The paper referred to by the commenter deals with a synthetic spheres dataset which we believe is not applicable to the use of GANs. Our focus is on real-life datasets collected from real examples. Furthermore, due to the recentness of the paper, we have not had the time to analyze it in detail.

---

### Author Response · Authors · 2018-01-05
**Revision**

We have posted a revision with an additional Appendix (F) for new white-box experiments on the CelebA dataset, as well as minor changes to the text.

---

### Decision · Program_Chairs · 2018-01-29
**ICLR 2018 Conference Acceptance Decision**

**Decision:**

Accept (Poster)

**Comment:**

The paper studied defenses against adversarial examples by training a GAN and, at inference time, finding the GAN-generated sample that is nearest to the (adversarial) input example. Next, it classifies the generated example rather than the input example. This defense is interesting and novel. The CelebA experiments the authors added in their revision suggest that the defense can be effective on high-resolution RGB images.